# Preparation of Fe_3_O_4_@polyoxometalates Nanocomposites and Their Efficient Adsorption of Cationic Dyes from Aqueous Solution

**DOI:** 10.3390/nano9040649

**Published:** 2019-04-23

**Authors:** Jie Li, Haiyan Zhao, Chenguang Ma, Qiuxia Han, Mingxue Li, Hongling Liu

**Affiliations:** Henan Key Laboratory of Polyoxometalates, Institute of Molecular and Crystal Engineering, College of Chemistry and Chemical Engineering, Henan University, Kaifeng 475004, China; hedalj@163.com (J.L.); 15890943739@163.com (H.Z.); 104753160773@vip.henu.edu.cn (C.M.); qiuxia_han@163.com (Q.H.)

**Keywords:** magnetic adsorbents, hybrids, nanocomposites, organic dyes

## Abstract

In this work, two magnetic adsorbents Fe_3_O_4_@**1** and Fe_3_O_4_@**2** were prepared by combining Fe_3_O_4_ nanoparticles and polyoxometalate hybrids [Ni(HL)_2_]_2_H_2_[P_2_Mo_5_O_23_]·4H_2_O (**1**), [H_2_L]_5_H[P_2_Mo_5_O_23_]·12H_2_O (**2**) (HL = 2-acetylpyridine-thiosemicarbazone). The temperature-dependent zero-field-cooled (ZFC) and field-cooled (FC) measurements indicated the blocking temperature at 160 K and 180 K, respectively. The Brunauer–Emmett–Teller (BET) surface area of Fe_3_O_4_@**1** and Fe_3_O_4_@**2** is 8.106 m^2^/g and 1.787 m^2^/g, respectively. Cationic dye methylene blue (MB) and anionic dye methyl orange (MO) were investigated for selective dye adsorption on Fe_3_O_4_@**1** and Fe_3_O_4_@**2**. The two adsorbents were beneficial for selective adsorption of cationic dyes. The adsorption efficiency of MB was 94.8% for Fe_3_O_4_@**1**, 97.67% for Fe_3_O_4_@**2**. Furthermore, the two adsorbents almost maintained the same adsorption efficiency after seven runs. The maximum MB adsorption capacity of Fe_3_O_4_@**1** and Fe_3_O_4_@**2** is 72.07 and 73.25 mg/g, respectively. The fourier transform infrared (FT-IR) and X-ray photoelectron spectroscopy (XPS) spectra of the adsorbents collected after adsorption of MB are very similar to the initial as-synthesized Fe_3_O_4_@polyoxometalates indicating the high stability of the two adsorbents. The adsorption kinetics indicated that the MB removal followed the pseudo-second-order model. These results showed that the two adsorbents had a potential application in treating wastewater.

## 1. Introduction

With industrial development, organic dyes have been one of the major water pollutants that cause serious environmental problems and threaten human health due to their toxicity [1,2]. Therefore, it is very urgent to develop advanced materials for dye removal. To date, two main strategies have been broadly explored. One strategy is photocatalytic degradation [3,4,5]. The process is an advanced oxidation, which is carried out under light irradiation. However, the activity of photocatalytic materials depends on their surface area, band gap, and electron pair generation to degrade dyes [6,7,8,9]. The other strategy is the adsorption process, which is recognized as the most simple, efficient and economic method for dye removal from water [10]. Activated carbon and silica, with large porous surface area, have been widely used in liquid-phase purification processes [11]. However, slow kinetics of adsorption, inadequate adsorption capacity, dye specificity and insufficient potential for recycling limit the use of traditional dye adsorbents.

Polyoxometalate hybrid systems provide a new pathway for combining multiple functional groups in a nano structure, which demonstrates great potential for application in selective adsorption [12]. Phosphomolybdates are a unique class of polyoxometalates, which have attracted increasing attention owing to its unique structure and potential application in the removal of dyes [13,14]. In particular, Strandberg-type phosphomolybdate [P_2_Mo_5_O_23_]^6−^ has been extensively studied. Thiosemicarbazones are of special application in efficient dye removal due to their amine and pyridyl, which are beneficial for the formation of hydrogen bonds and π–π stacking interactions [15]. Transition metal ions have been used for the removal of dyes due to coordination with most of organic dyes containing –C=C–, –N=N– and heterocyclic compounds [16]. Thus, it is possible to form a polyoxometalate hybrid to enhance the adsorption efficiency by combining [P_2_Mo_5_O_23_]^6−^ with nickel (II) and HL. Nevertheless, there is a problem separating them from water. In order to resolve this problem, it will be a good choice to load the complex on to the nanocarriers. Fe_3_O_4_ magnetic nanoparticles are widely explored due to their exceptional physicochemical properties, low cost, simple preparation process and easy isolation [17,18].

Herein, two nanocomposites, Fe_3_O_4_@**1** and Fe_3_O_4_@**2** were synthesized by combining Fe_3_O_4_ nanoparticles and pre-synthesized polyoxometalate hybrids **1**, **2**. Fe_3_O_4_@**1** and Fe_3_O_4_@**2** were explored in the adsorption of methylene blue (MB) and methyl orange (MO), and it turned out that the two nanocomposites had high efficiency for selective adsorption of cationic dyes. Furthermore, the stability and recyclability of Fe_3_O_4_@**1** and Fe_3_O_4_@**2** were also studied. The two nanocomposites could be isolated easily from the sample solution by applying an external magnetic field. These results proved that the two magnetic nanocomposites could be of interest in treating wastewater.

## 2. Experiment Section

### 2.1. Materials and Measurements

All chemicals were used as purchased without purification. The Fe_3_O_4_ nanoparticles were synthesized as previously described [19]. The structures and grain sizes of two nanocomposites were analyzed by X-ray diffraction (XRD, X’Pert Pro, Bruker, Karlsruhe, Germany) using Cu *Kα* radiation and transmission electron microscopy (TEM, JEOL 2010F, JEOL Ltd., Tokyo, Japan) including the mode of high resolution (HRTEM). Elemental analyses (C, H and N) were implemented on a Flash 2000 analyzer (Elementar, Hessia, Germany). Inductively coupled plasma analysis was performed on an Optima 2100 DV (PerkinElemer, Waltham, MA, USA). The fourier transform infrared (FT-IR) spectra were measured on a VERTEX 70 (Bruker, Karlsruhe, Germany) using KBr pellets in the range of 4000–500 cm^−1^. The ultraviolet–visible (UV–vis) absorption spectra were obtained with a TU–1900 spectrometer (Beijing Purkinje General Instrument Co., Ltd., Beijing, China) at room temperature. The magnetic property was investigated by a vibrating sample magnetometer (VSM) and physical property measurement system (PPMS) (Lakeshore 7300, Quantum Design, San Diego, CA, USA). X-ray photoelectron spectroscopy (XPS) was observed on a Thermo ESCALAB 250XI photoelectron spectrometer with Al *Kα* X-ray as the excitation source (ThermoFisher Scientific, Waltham, MA, USA). The N_2_ absorption-desorption isotherms for determining the Brunauer–Emmett–Teller (BET) surface area were obtained with an ASAP 2460 Micromeritics instrument (Micromeritics, Norcross, Georgia) at 77 K.

### 2.2. Preparation of [Ni(HL)_2_]_2_H_2_[P_2_Mo_5_O_23_]·4H_2_O *(**1**)*

A mixture of Ni(ClO_4_)_2_·6H_2_O (0.093 g, 0.25 mmol), HL (0.098 g, 0.5 mmol), methanol (10 mL) and H_2_O (15 mL) was stirred at 60 °C for 30 min. After cooling to room temperature, a 10 mL aqueous solution of Na_2_MoO_4_·2H_2_O (0.242 g, 1.0 mmol) was added, then the pH value was adjusted from 6.57 to 3.0 with H_3_PO_4_ (85%) under continuous stirring. The mixture was stirred for another 30 min at 60 °C, cooled and filtered. The filtrate was allowed to stand for slow evaporation at room temperature. Brown blocky crystals of **1** were collected in about 63.9% yield (based on Ni) after five days. Elemental analysis for C_32_H_50_Mo_5_N_16_Ni_2_O_27_P_2_S_4_: calcd (%): C 20.46, H 2.68, Mo 25.54, N 11.93, Ni 6.25, P 3.30; Found (%): C 20.56, H 2.60, Mo 25.74, N 12.01, Ni 6.45, P 3.37. IR (KBr, cm^−1^): 3442 (w), 3278 (w), 3165 (w), 1615 (s), 1477 (w), 1377 (s), 1315 (w), 1187(w), 1107(w), 1056 (m), 928 (s), 873 (w), 780 (w), 703 (s).

### 2.3. Preparation of [H_2_L]_5_H[P_2_Mo_5_O_23_]·12H_2_O *(**2**)*

A 25 mL solution (V_methanol_/V_water_ = 2/3) containing HL (0.098 g, 0.5 mmol) was stirred at 60 °C for 30 min. After cooling to room temperature, a 10 mL aqueous solution of Na_2_MoO_4_·2H_2_O (0.242 g, 1.0 mmol) was added, then the pH value was adjusted from 7.48 to 3.0 with H_3_PO_4_ (85%) under continuous stirring. The mixture was stirred for another 30 min at 60 °C, cooled and filtered. The filtrate was allowed to stand for slow evaporation at room temperature. After three days grass green crystals of **2** were collected in about 53.6% yield (based on Mo). Elemental analysis for C_40_H_80_Mo_5_N_20_O_35_P_2_S_5_: calcd (%): C 22.84, H 3.83, Mo 22.81, N 13.32, P 2.95; Found (%): C 22.94, H 3.78, Mo 22.68, N 13.77, P 2.88. IR (KBr, cm^−1^): 3400 (w), 3272 (w), 3173 (w), 1621 (s), 1454 (w), 1370 (w), 1245 (m), 1151(m), 917 (s), 837 (w), 693 (s).

### 2.4. Preparation of Nanocomposites *Fe_3_O_4_@**1*** and *Fe_3_O_4_@**2***

Fe_3_O_4_@**1** was synthesized by ultrasonic method. A mixture of reactants was obtained by adding **1** (50 mg) and Fe_3_O_4_ (7.5 mg) into a 25 mL beaker containing water (5 mL) and ethanol (5 mL). After ultrasound for 10 h, a uniform and turbid liquid was obtained. Then the resulting products were collected using a magnet putting on one side of the beaker to separate them from the turbid liquid. The production was washed with water several times. The weight ratio of **1** in Fe_3_O_4_@**1** was 96.68% (Appendix A).

The preparation process of Fe_3_O_4_@**2** was similar to Fe_3_O_4_@**1** except that **1** (50 mg) was replaced by **2** (50 mg). The weight ratio of **2** in Fe_3_O_4_@**2** was 96.50% (Appendix A).

### 2.5. X-ray Crystallographic Study

The suitable single crystals of two hybrids **1**–**2** were selected and encapsulated in two glass tubes, respectively, then transferred to a X-ray single crystal diffractometer (SMART APEX-II CCD) at 296(2) K. Routine Lorentz and polarization corrections were used. Absorption correction was on the basis of multiple and symmetry equivalent reflections in data set with the SADABS program. The structures were solved by direct methods and refined by full-matrix least-squares on *F^2^* using the SHELXL program package [20,21]. No hydrogen atoms related to the water molecules were located from the disparate Fourier map. The crystallographic data and structure refinements for hybrids **1**–**2** were summarized in Table 1. The CCDC number of **1** and **2** is 1872074 and 1872075, respectively.

### 2.6. Adsorption Experiments

The typical adsorption process is as follows: the adsorbent (25.0 mg) were added to the MB aqueous solution (15 mL) at a concentration of 15 mg/L. The solutions were magnetically stirred (800 rpm) at room temperature in the dark. At several time intervals, the 4 mL solution was removed and analyzed using a UV–vis spectrophotometer at the calibrated maximum wavelength of 664 nm. The supernatant was collected at different time intervals for a kinetic study. The percentages of MB removal and equilibrium adsorption capacity (*q_e_*) of the two nanocomposites were calculated using Equations (1) and (2) as follows [22,23]:(1)$$\mathrm{Dye}\;\mathrm{removal}\;\left(\%\right)=\frac{C_{0}-C_{t}}{C_{0}}=\frac{A_{0}-A_{t}}{A_{0}}$$
(2)$$q_{e}=\left(\frac{C_{0}-C_{e}}{W}\right)\times V$$
where *C_0_* and *C_e_* are the initial and equilibrium concentrations of MB in the water, *C_t_* is the concentration of MB at any specified time t, *A_0_* and *A_t_* represent the initial and the time t absorbance of MB at 664 nm, *V* is the volume of MB solution and *W* is the mass of adsorbent.

## 3. Results and Discussion

### 3.1. Crystal Structure Descriptions of Hybrids ***1**–**2***

Both hybrids **1** and **2** consist of a typical Strandberg-type [P_2_Mo_5_O_23_]^6−^ unit [24]. [P_2_Mo_5_O_23_]^6−^ comprises two tetrahedral {PO_4_} and five octahedral {MoO_6_}, which can be seen as a pentagonal ring formed by octahedra in edge- and corner-sharing mode. In hybrid **1** (Figure 1a), there are a [P_2_Mo_5_O_23_]^6−^ unit, two [Ni(HL)_2_]^2+^ cations, two protons and four crystal water molecules. The anion [P_2_Mo_5_O_23_]^6−^ and cationic [Ni(HL)_2_]^2+^ are combined by electrostatic interaction and hydrogen bonds (dotted light orange lines in Figure 1a). In [Ni(HL)_2_]^2+^ cations, each Ni(II) is coordinated by two S atoms and four N atoms from two ligands HL (Figure 1d). The bond lengths ranges of Ni–S and Ni–N are 2.377(2)–2.43(1) and 2.009(2)–2.100(2), respectively. The ligand HL contains primary amine, secondary amine and pyridine groups (Figure 1c), which are beneficial for the formation of hydrogen bonds and π–π stacking interactions with dyes. In addition, the hydrogen bonds involved the terminal nitrogen atoms with the oxygen atoms of [P_2_Mo_5_O_23_]^6−^ were observed. The bond lengths and bond angles of N(1)–H(1B)···O(12), N(2)–H(2B)···O(10) are 3.083(17), 2.754(16) and 145.5(8)°, 165.4(7)°, respectively. Then the molecules generate a three-dimensional network with hydrogen bonds and π–π stacking interactions (Figure 1b). As a reference, hybrid **2** was synthesized, in which the metal ions are excluded. **2** consists of five [H_2_L]^+^, a [P_2_Mo_5_O_23_]^6−^ anions, one proton and 12 crystal water molecules. Figure 1e exhibits the polyhedral/wire-stick representation of the 3D network of hybrid **2**. The coexistence of a large amount of negative charge on the surface of [P_2_Mo_5_O_23_]^6−^ and amino-incorporated ligands will provide a way to accelerate the selective adsorption of cationic organic dyes by hydrogen bonds and π–π stacking interactions. 

### 3.2. Fourier Transform Infrared (FT-IR) Spectroscopy

The fourier transform infrared (FT-IR) spectra were recorded in the range of 4000–500 cm^−1^ for **1**, Fe_3_O_4_@**1**, **2**, Fe_3_O_4_@**2** and Fe_3_O_4_ (Figure 2). The broad peak at 3429 cm^−1^ is associated with the covalent O–H stretching vibrations of water molecules [25]. The bands in the range 3278–3164 cm^−1^ are associated with the stretching vibrations of the *ν*(N–H) bonds. The peaks between 1620 and 1561 cm^−1^ are attributed to the presence of *ν*(C=N) bonds [26]. The scope of the characteristic peak of the P–O band is in the range of 1051–1121 cm^−1^. Additionally, the strong bands between 672 and 960 cm^−1^ are attributed to the *ν*(Mo–O_b_) and *ν*(Mo–O_t_) modes of the [P_2_Mo_5_O_23_]^6−^ [27]. The IR spectra of Fe_3_O_4_@**1**, Fe_3_O_4_@**2** and Fe_3_O_4_ all exhibit the same characteristic peaks at *ca* 590 cm^−1^ assigned to the Fe–O, indicating the presence of Fe_3_O_4_. 

### 3.3. Ultraviolet–visible (UV–vis) Spectroscopy

As shown in Figure 3, the UV–vis spectra of **1** and Fe_3_O_4_@**1** show the same absorption peaks in aqueous solution at 212 and 262 nm, **2** and Fe_3_O_4_@**2** exhibit the same adsorption peaks at 208 and 306 nm, which are all attributed to O_t_→Mo and O_b_→Mo charge-transfer transition [28]. As a comparison, no absorption peak is observed for Fe_3_O_4_ nanoparticles. 

### 3.4. Magnetic Properties of Fe_3_O_4_@1 and Fe_3_O_4_@2

The magnetic properties of Fe_3_O_4_@**1** and Fe_3_O_4_@**2** were investigated by VSM and PPMS. Figure 4a represents the hysteresis curves of Fe_3_O_4_@**1,** Fe_3_O_4_@**2** and Fe_3_O_4_. At 300 K, Fe_3_O_4_ shows a coercivity tend to ~23 Oe and the magnetization of ~68 emu/g under ~30,000 Oe, while Fe_3_O_4_@**1** and Fe_3_O_4_@**2** show the coercivity tend to ~32 Oe and ~47 Oe, the magnetization of ~11 emu/g and ~10 emu/g, respectively. The magnetization decreases due to the combination of the polyoxometalate on the surface of Fe_3_O_4_ nanoparticles [29]. At 5 K, Fe_3_O_4_@**1** and Fe_3_O_4_@**2** show the coercivity tend to ~240 Oe and ~255 Oe, the magnetization of ~14.4 emu/g and ~14.7 emu/g under ∼10,000 Oe, respectivey (Figure 4b). The coercivity increases with decreasing temperature, which is the result of a thermal effect. The hysteresis curves of Fe_3_O_4_@**1** and Fe_3_O_4_@**2** indicate that Fe_3_O_4_@**1** and Fe_3_O_4_@**2** behave soft ferromagnetic at 300 K and become ferromagnetic at 5 K, with a significant enhancement in magnetization as the temperature decreases. 

Figure 4c,d subsequently show the magnetization-temperature curves in two modes of field-cooled (FC) and zero-field-cooled (ZFC) in the temperature range of 2–300 K under a given applied magnetic field of 500 Oe, revealing the superparamagnetic response of Fe_3_O_4_@**1** and Fe_3_O_4_@**2**. It should be mentioned that the two samples exhibited quite similar thermal relaxation behavior of nanoparticles’ magnetic moment. The average blocking temperature is estimated to be ~160 K and ~180 K for Fe_3_O_4_@**1** and Fe_3_O_4_@**2**, respectively [30]. Below the blocking temperature, the ZFC magnetization falls sharply down to 2 K whereas FC magnetization decreases very little.

### 3.5. Separation and Redispersion Process of *Fe_3_O_4_@**1*** and *Fe_3_O_4_@**2***

As shown in Figure 5a, under the influence of an external magnetic field, Fe_3_O_4_@**1** in water rapidly changed from a yellow homogeneous dispersion to colourless transparent solution. The collected nanocomposites can be easily and reversibly dispersed by agitation after removing the magnetic field and the above process can be repeated many times. A similar process happens to Fe_3_O_4_@**2** (Figure 5b). The aggregation and dispersion experiments show that there is no compound left after the two nanocomposites are collected by magnet, which indicates that polyoxometalates hybrids and Fe_3_O_4_ are successfully combined together as the hybrids **1** and **2** are non-magnetic. Furthermore, these experiments also suggest that Fe_3_O_4_@**1** and Fe_3_O_4_@**2** possess strong magnetic responsiveness and redispersibility, which are distinct advantageous to their adsorption application.

### 3.6. The Morphology and Particle Size Distribution of *Fe_3_O_4_@**1*** and *Fe_3_O_4_@**2***

In order to investigate the morphology, particle sizes and size distributions of the two nanocomposites, TEM, HRTEM and XRD experiments were tested. As shown in Figure 6a,d, the two nanocomposites are high levels of crystallization, virtually uniform and nearly round in shape. The size distributions of two nanocomposites were obtained from the size counting of a series of TEM photos, in which the histograms showed average size of 20.70 nm and 20.40 nm corresponding to Fe_3_O_4_@**1** and Fe_3_O_4_@**2**, respectively. The distribution of the two nanocomposites can be reasonably described by the curve of Gaussian function (Figure 6b,e). Figure 6c,f exhibit the HRTEM images of two nanocomposites, the clear crystal lattices of Fe_3_O_4_@**1** and Fe_3_O_4_@**2** correspond to the (311) and (220) reflection of the Fe_3_O_4_, respectively. Meantime, it can be found that an obvious interface line (marked with white line) delimitate the regions of Fe_3_O_4_@**1** and Fe_3_O_4_@**2**. The HRTEM images indicate that parts of polyoxometalates are successfully covered in the surface of Fe_3_O_4_ [31], which are in agreement with the aggregation and dispersion experiments. 

XRD patterns of simulation of **1** and **2**, Fe_3_O_4_@**1**, Fe_3_O_4_@**2** and Fe_3_O_4_ are listed in Figure 7a. Fe_3_O_4_ exhibited a round structure that fitted to JCPDS No. 75-0449. The peaks at 30.43°, 35.7°, 43.3°, 53.7°, 57.3°, 62.9° and 74.3° (as labeled by the asterisk) are corresponding to the (220), (311), (400), (422), (511), (440) and (533) plans of the Fe_3_O_4_, respectively. The average particle sizes of Fe_3_O_4_@**1** and Fe_3_O_4_@**2** were 20.3 nm and 19.6 nm, which were evaluated by Debye-Scherrer equation. The result of XRD analysis agrees well with that of TEM analysis. The simulated peak positions of **1** and **2** match with those of Fe_3_O_4_@**1** and Fe_3_O_4_@**2**, indicating that nanocomposites Fe_3_O_4_@**1** and Fe_3_O_4_@**2** were successfully synthesized. The specific surface area of the samples was confirmed by N_2_ adsorption-desorption isotherms. It turns out that the BET surface area of Fe_3_O_4_@**1** and Fe_3_O_4_@**2** are 8.106 m^2^/g and 1.787 m^2^/g, respectively. Figure 7b shows the N_2_ adsorption-desorption isotherms of the Fe_3_O_4_@**1** sample.

### 3.7. Dye Adsorption Properties

In recent decades, much attention has been devoted to adsorb organic dyes from wastewater with Fe_3_O_4_-based nanocomposites. The adsorption activities of Fe_3_O_4_@**1** and Fe_3_O_4_@**2** were explored by removing cationic dye MB and anionic dye MO from wastewater. As exhibited in Figure 8a,b, Fe_3_O_4_@**1** and Fe_3_O_4_@**2** displayed interesting adsorbent properties for cationic dye MB and little absorption activity for anionic dye MO. The adsorption efficiency of MB was 94.8% for Fe_3_O_4_@**1** in 240 min, 97.67% for Fe_3_O_4_@**2** in 60 min. However, the adsorption efficiency of MO was only 13.13% for Fe_3_O_4_@**1** and 8.84% for Fe_3_O_4_@**2**. The maximum MB adsorption capacity of Fe_3_O_4_@**1** and Fe_3_O_4_@**2** is 72.07 and 73.25 mg/g, respectively. In comparison with the other Fe_3_O_4_-based adsorbents Fe_3_O_4_/PPC MPs [32], our prepared adsorbents displayed a higher adsorption capacity. Due to the similar size of MB and MO (Figure 8c,d), the selective adsorption for MB might be attributed to the electrostatic interactions between adsorbents and cationic dye molecules.

To confirm this hypothesis, adsorption kinetic studies of the removal of MB were carried out. As a proof-of-concept experiment, only the adsorption activity of Fe_3_O_4_@**1** was systematically studied as an example. In order to explore the adsorption kinetics precisely, 10 mg/L, 20 mg/L, 25 mg/L and 30 mg/L of MB were prepared for parallel experiments. A pseudo-first-order model and pseudo-second-order kinetic model [22,33] have been used to fit the experimental data of Fe_3_O_4_@**1**.
(3)$$\log\left(q_{e}-q_{t}\right)=\log\;q_{e}-k_{1}t$$
(4)$$\frac{t}{q_{t}}=\frac{1}{k_{2}{\left(q_{e}\right)}^{2}}+\frac{t}{q_{e}}$$
where *k*_1_ is the rate constant of pseudo-first-order, *k*_2_ is the rate constant of pseudo-second-order, *q_t_* is the amount of adsorbed MB at time *t*. 

As expected, the values of correlation coefficient (*R^2^*) described the pseudo-second-order model very well (Table 2, Figure 9a), indicating that the adsorption process of Fe_3_O_4_@**1** toward MB obeyed the pseudo-second-order kinetics. These results strongly demonstrated that the adsorption capacity of Fe_3_O_4_@**1** was due to the electron transfer [34]. In order to further confirm the hypothesis, the other cationic dye rhodamine B (RhB) was also used to test the selective adsorption of Fe_3_O_4_@**1**, which turned out that the adsorption efficiency of RhB was 95.98% in 240 min (Figure 9b). 

The two isotherms of Langmuir and Freundlich were used to further study the adsorption experimental data. Thus a linear equation for each model of the two isotherms was introduced to confirm the equilibrium characteristics of adsorption between Fe_3_O_4_@**1** and MB. The two isotherm equations are listed as follows [35]: (5)$$\frac{C_{e}}{q_{e}}=\frac{C_{e}}{q_{m}}+\frac{1}{K_{L}q_{m}}$$
(6)$$lgq_{e}=lgK_{f}+\left(\frac{1}{n}\right)lgC_{e}\;$$
where *q_max_* is the maximum adsorption capacity. *Ce* is the equilibrium concentration of MB in the solution. *K_L_* and *K_F_* are the constant of the Langmuir and Freundlich model. *n* is the adsorption intensity. 

The adsorption isotherms of MB are shown in Figure 10, and their detailed parameters are exhibited in Table 3. The Langmuir and Freundlich isotherms fitted the experimental data for Fe_3_O_4_@**1** with *R^2^* values of 0.9968 and 0.9977, respectively. These results indicated that the Langmuir and Freundlich isotherm provided the highest fit to the adsorption process. 

The effect of initial pH values on MB adsorption over the Fe_3_O_4_@**1** was studied by adjusting the solution pH to 3, 5, 7, 9, 11 and 13 (Figure 11a). It can be found that the initial pH is important for the adsorption of MB. When the initial pH values were in the range of 3~7, the adsorption equilibrium amount of Fe_3_O_4_@**1** shows an obvious increase. However, it dropped sharply when the initial pH values ranging from 9 to 13 because Fe_3_O_4_@**1** will decompose above pH 9. The MB adsorption capability retained a relatively high condition when the initial pH values were ca. 7, which was used in the experiments. 

For comparison with Fe_3_O_4_@**1**, effect of blank, Fe_3_O_4_, **1**, Fe_3_O_4_ and **1** were tested, respectively. (Figure 11b). Blank experiment indicated the MB aqueous solution hardly changed after 4 h. Under the same condition, the adsorption efficiency of Fe_3_O_4_, **1**, Fe_3_O_4_ and **1**, Fe_3_O_4_@**1** was 19.70%, 56.66%, 63.1%, 94.80%, respectively. Thus, the active site of adsorption reaction of Fe_3_O_4_@**1** might concentrate on polyoxometalate hybrid **1** and the Fe_3_O_4_@**1** showed a better adsorption performance when Fe_3_O_4_ and **1** were bound together.

The reusability and stability of adsorbents are very important factors for their practical application [36,37]. The reusability of the two adsorbents was studied via seven repeated experiments under constant experimental conditions (initial MB concentration of 15 mg/L). For this purpose, the adsorbents were collected by a magnet putting on one side of the beaker, and washed with ethanol in an ultrasonic environment. These processes were repeated seven times. After the fifth washing, the collected adsorbents were dried in an oven at 60 °C. Then the recovered adsorbents were used in another adsorption run. The results exhibited that the adsorption capacities decreased only about 3.73% for Fe_3_O_4_@**1** and 3.5% for Fe_3_O_4_@**2** after seven runs (Figure 12a,b), indicating that the two nanocomposites could be reused as efficient adsorbents for wastewater. During recycling, the two adsorbents almost maintained the same adsorption capacity. The slight decline might correspond to the loss of adsorbents for washing and FT-IR testing. The FT-IR spectra of the adsorbents collected after adsorption of MB are very similar to that observed for the initial as-synthesized (Figure 12c,d). Moreover, in order to further investigate the stability of Fe_3_O_4_@**1** and Fe_3_O_4_@**2**, the XPS of as-synthesized and after-adsorption were tested (Figure 12e,f). These results clearly indicated the high stability of the two adsorbents. It should be noted that the XPS of as-synthesized nanocomposites indicated that Fe_3_O_4_ and **1**, **2** were successfully bound together again. Consequently, Fe_3_O_4_@**1** and Fe_3_O_4_@**2** were almost restored to its original adsorption property and showed excellent stability, which indicated a great possibility for water treatment.

## 4. Conclusions

In conclusion, two magnetic adsorbents Fe_3_O_4_@**1** and Fe_3_O_4_@**2** were synthesized by the ultrasonic method with Fe_3_O_4_ and polyoxometalate hybrids **1**, **2**. The ZFC and FC measurements indicated the blocking temperatures at 160 K and 180 K, respectively. The BET surface areas of Fe_3_O_4_@**1** and Fe_3_O_4_@**2** were 8.106 m^2^/g and 1.787 m^2^/g, respectively. The two adsorbents were highly efficient for selective removal of cationic dyes from aqueous solution. The adsorption efficiency of MB solution was 94.8% for Fe_3_O_4_@**1** and 97.67% for Fe_3_O_4_@**2**. The maximum MB adsorption capacities of Fe_3_O_4_@**1** and Fe_3_O_4_@**2** were 72.07 and 73.25 mg/g, respectively. The adsorption kinetics indicated that the MB removal followed the pseudo-second-order model. More importantly, the two adsorbents exhibited good recyclability and high stability. After seven cycles, the two adsorbents almost maintained the same adsorption capacity. The FT-IR and XPS spectra of the two adsorbents collected from the adsorption experiments agreed well with the fresh samples indicating their high stability. These results clearly indicate that the two nanocomposites have potential applications in wastewater treatment.

## Figures and Tables

**Figure 1 nanomaterials-09-00649-f001:**
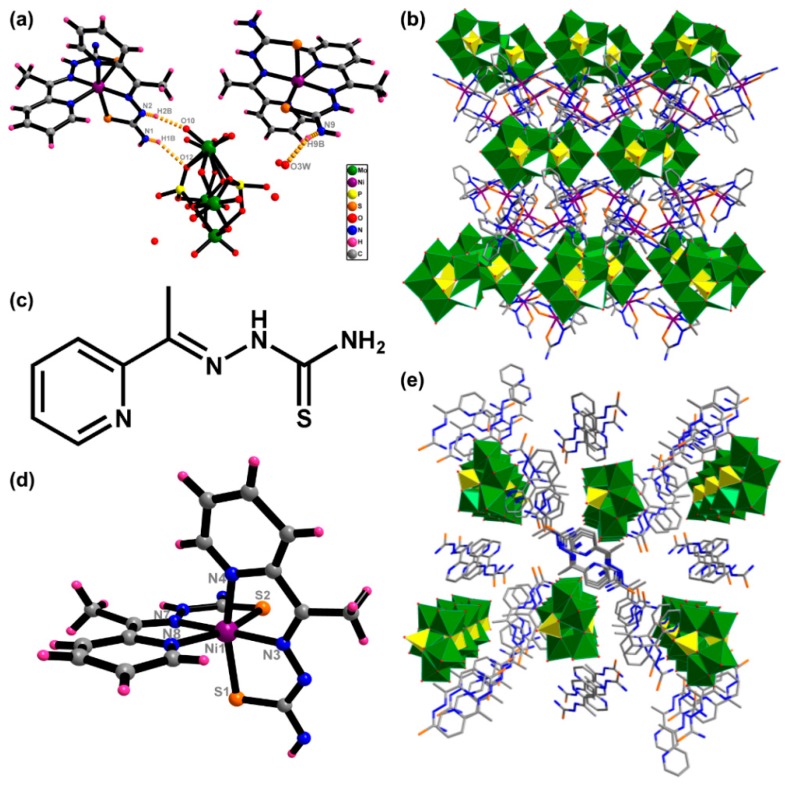
The structure of (**a**) **1** and (**c**) HL. Coordination patterns of (**d**) Ni^2+^ ion. Polyhedral/wire-stick representation of the 3D network of (**b**) **1** and (**e**) **2**.

**Figure 2 nanomaterials-09-00649-f002:**
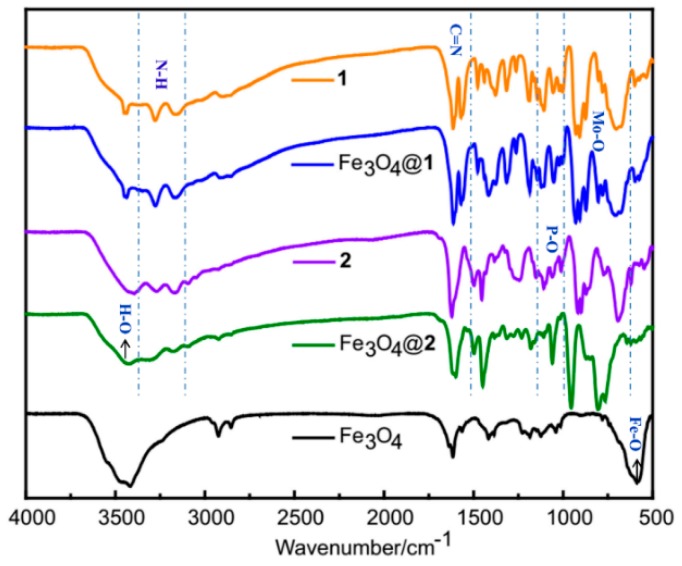
The fourier transform infrared (FT-IR) spectra of **1**, Fe_3_O_4_@**1**, **2**, Fe_3_O_4_@**2** and Fe_3_O_4_.

**Figure 3 nanomaterials-09-00649-f003:**
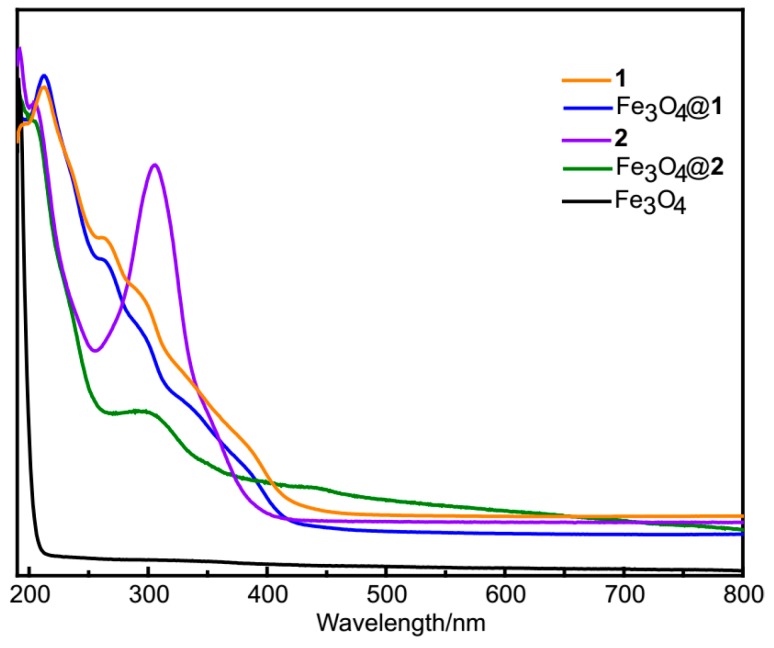
The *Ultraviolet–**visible) (UV–**vis)* spectra of **1**, Fe_3_O_4_@**1**, **2**, Fe_3_O_4_@**2** and Fe_3_O_4_.

**Figure 4 nanomaterials-09-00649-f004:**
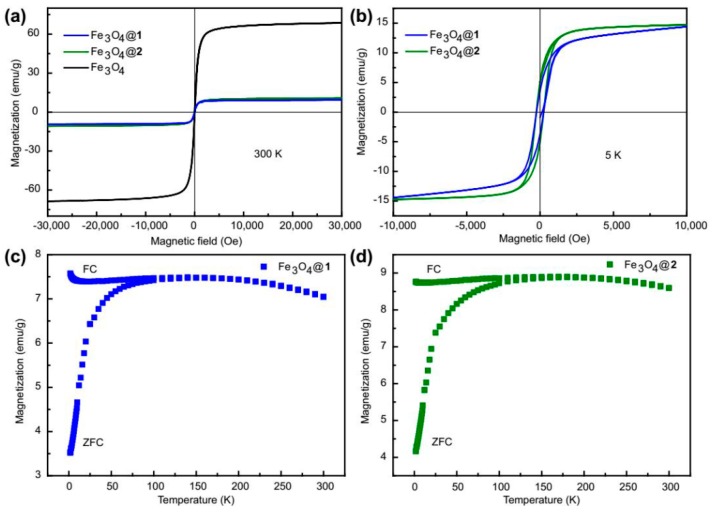
Magnetic measurements. Hysteresis curves of Fe_3_O_4_@**1**, Fe_3_O_4_@**2** and Fe_3_O_4_ recorded at (**a**) 300 K and (**b**) 5 K. FC and ZFC curves of (**c**) Fe_3_O_4_@**1** and (**d**) Fe_3_O_4_@**2** under the magnetic field of 500 Oe.

**Figure 5 nanomaterials-09-00649-f005:**
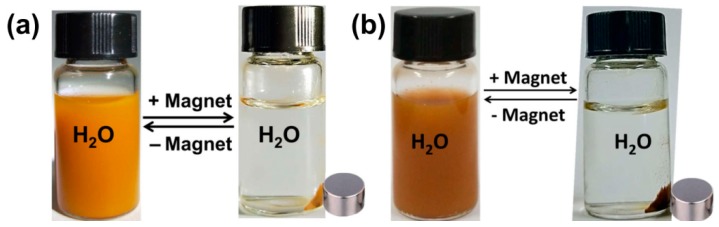
The dispersion-collection process of (**a**) Fe_3_O_4_@**1** and (**b**) Fe_3_O_4_@**2**.

**Figure 6 nanomaterials-09-00649-f006:**
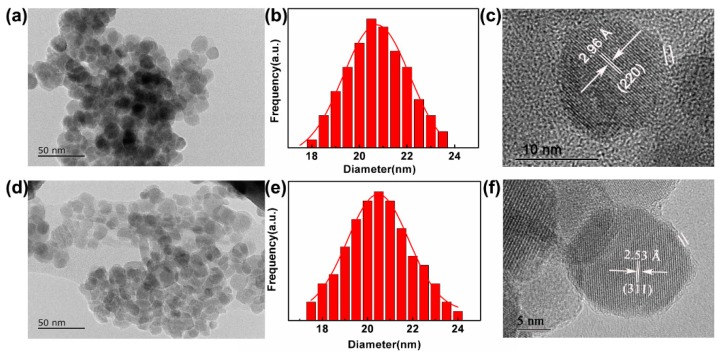
Transmission electron microscope (TEM) images of (**a**) Fe_3_O_4_@**1**, (**d**) Fe_3_O_4_@**2**. Particle size histogram with Gaussian fit of (**b**) Fe_3_O_4_@**1**, (**e**) Fe_3_O_4_@**2**. HRTEM of (**c**) Fe_3_O_4_@**1**, (**f**) Fe_3_O_4_@**2**.

**Figure 7 nanomaterials-09-00649-f007:**
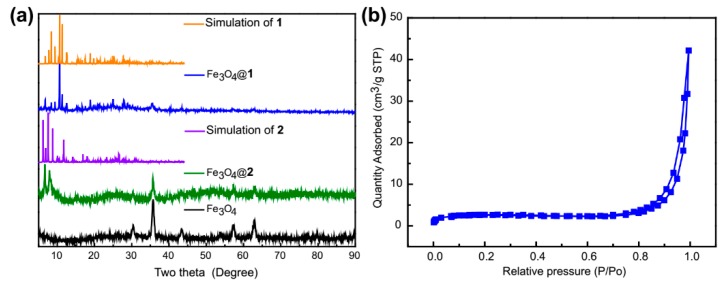
(**a**) The X-ray diffraction (XRD) analyses of simulation of **1** and **2**, Fe_3_O_4_@**1**, Fe_3_O_4_@**2** and Fe_3_O_4_. (**b**) the N_2_ adsorption-desorption isotherms of Fe_3_O_4_@**1** sample.

**Figure 8 nanomaterials-09-00649-f008:**
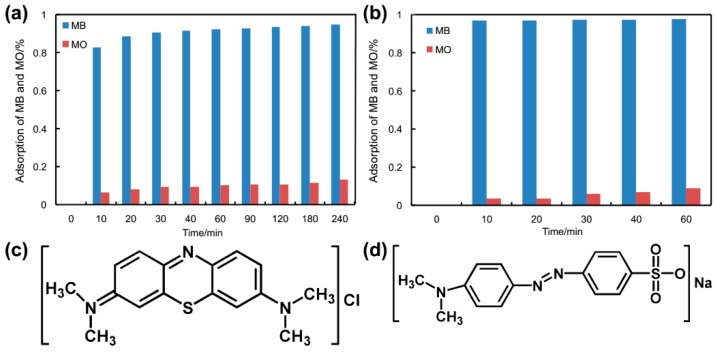
Absorption of MB and MO aqueous solution in the presence of (**a**) Fe_3_O_4_@**1**, (**b**) Fe_3_O_4_@**2** and the chemical structure of (**c**) MB, (**d**) MO.

**Figure 9 nanomaterials-09-00649-f009:**
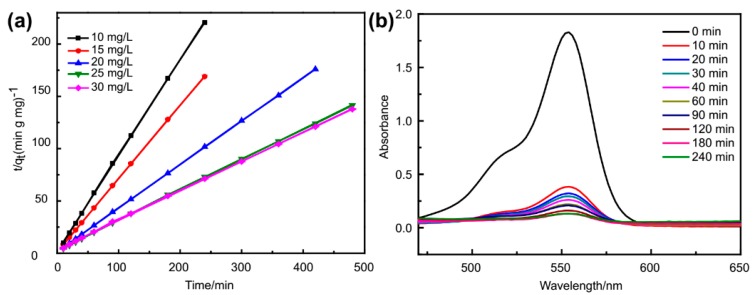
(**a**) Plots of pseudo-second-order kinetics for the adsorption of methylene blue (MB) over adsorbent Fe_3_O_4_@**1**; (**b**) absorption spectra of rhodamine B (RhB) aqueous solution in the presence of Fe_3_O_4_@**1**.

**Figure 10 nanomaterials-09-00649-f010:**
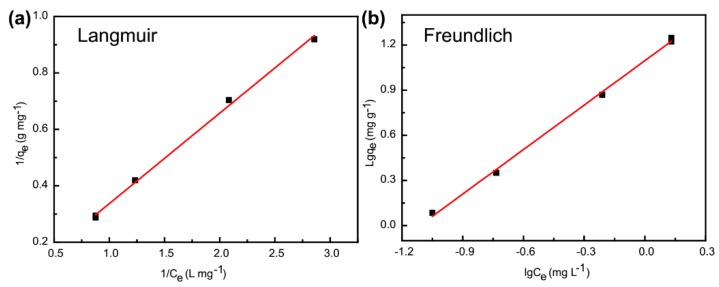
Isotherms of (**a**) Langmuir and (**b**) Freundlich model of MB adsorption on the Fe_3_O_4_@**1**.

**Figure 11 nanomaterials-09-00649-f011:**
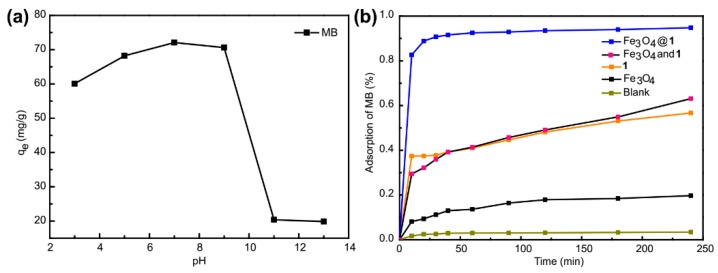
(**a**) Effect of pH on MB adsorption over the Fe_3_O_4_@**1** (initial solutions, 15 mg L^−1^; temperature, 298 K). (**b**) Adsorption activity comparison of blank, Fe_3_O_4_, **1**, Fe_3_O_4_ and **1**, Fe_3_O_4_@**1**.

**Figure 12 nanomaterials-09-00649-f012:**
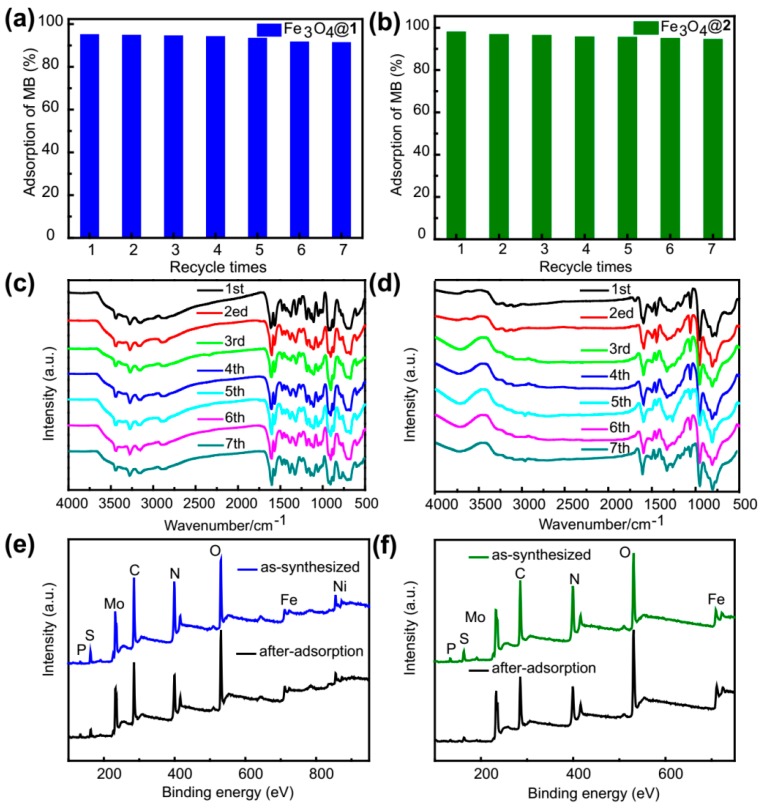
Reusability studies of (**a**) Fe_3_O_4_@**1** and (**b**) Fe_3_O_4_@**2**. The FT-IR spectra of (**c**) Fe_3_O_4_@**1** and (**d**) Fe_3_O_4_@**2** for adsorption of MB. The as-synthesized and after-adsorption X-ray photoelectron spectroscopy (XPS) of (**e**) Fe_3_O_4_@**1** and (**f**) Fe_3_O_4_@**2**.

**Table 1 nanomaterials-09-00649-t001:** Summary of crystal data and refinement results for hybrids **1** and **2.**

Hybrids	1	2
Empirical formula	C_32_H_50_Ni_2_Mo_5_N_16_O_27_P_2_S_4_	C_40_H_80_Mo_5_N_20_O_35_P_2_S_5_
Formula weight	1878.13	2103.15
Temperature	296(2)	296(2)
Crystal system	Monoclinic	Monoclinic
space group	*C/c*	*P2(1)/c*
*a*/[Å]	29.786(10)	10.8531(6)
*b*/[Å]	12.676(4)	28.4792(16)
*c*/[Å]	19.097(7)	25.8460(15)
β[°]	119.353(5)	96.9100(10)
Z	4	4
Volume/[Å^3^]	6285.0(4)	7930.7(8)
Calculated density/[g·cm^−3^]	1.972	1.745
*μ*/[mm^−1^]	1.825	1.034
F(000)	3672	4160
Crystal size/mm^3^	0.21 × 0.18 × 0.17	0.23 × 0.23 × 0.20
Theta range for data collection	2.16–25.00	1.59–25.00
Limiting indices	−35 ≤ h ≤ 32, −15 ≤ k ≤ 12, −21 ≤ l ≤ 22	−12 ≤ h ≤ 12, −30 ≤ k ≤ 33, −27 ≤ l ≤ 30
Data/restraints/parameters	7788/398/788	13927/0/959
Reflections collected/unique	12,785/7788 [R(int) = 0.0261]	40,316/13,927 [R(int) = 0.0328]
Goodness-of-fit on F^2^	1.100	1.020
Final R indices [*I ≥ 2σ(I)*]	0.0611, 0.1596	0.0468, 0.1451
R indices (all data)	0.1053, 0.2299	0.0610, 0.1539
Largest diff. peak and hole/[e·Å^−3^]	2.251, −1.288	2.951, −1.150

**Table 2 nanomaterials-09-00649-t002:** Fitting parameters (*R^2^*) by pseudo-second-order models.

MB (mg/L)	10	15	20	25	30
*R^2^*	0.99978	0.99996	0.99996	0.99979	0.9997

**Table 3 nanomaterials-09-00649-t003:** Parameters of isothermal for the adsorption of MB on the Fe_3_O_4_@**1**.

Equations	Parameters		
Langmuir	*q_m_* (mg g^−1^)	*K_L_*	*R^2^*
	71.0	19.53	0.9968
Freundlich	*n*	*K_F_*	*R^2^*
	0.90	17.82	0.9977

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
