# Peer review of "Preparation of Fe3O4@polyoxometalates Nanocomposites and Their Efficient Adsorption of Cationic Dyes from Aqueous Solution"

_nanomaterials, 2019, doi:10.3390/nano9040649_

Round 1
Reviewer 1 Report
I think, the manuscript has been well improved after the incorporation of the suggested revision. It can now be considered for publication.
Author Response
Thank you very much for your kind work and consideration on publication of our paper. On behalf of my co-authors, we would like to express our great appreciation to you.

Reviewer 2 Report
This manuscript by Li et al. reports on the use of polyoxometalate-hybrids@iron oxide nanocomposites for the removal of cationic dyes in aqueous solution. POM-hybrids based on Strandberg-type phosphomolybdates and organic molecules have been characterized by single-crystal X-ray diffraction , whereas the formation of composites have been analyzed by IR and UV-Vis spectroscopies, magnetic measurements and TEM images. Although results from dye absorption studies could be relevant enough to recommend the publication of this work in Nanomaterials, I cannot accept it in its current form, because there are major inconsistencies and inaccuracies all along the manuscript. Therefore, I would suggest a deep revision before reconsideration based on the following points:
1) The nature of the composite is my main concern. Are they core-shell particles? I do not think so, because TEM images show homogeneous phases of Fe3O4. There is no presence of darker points corresponding to POM units with higher electronic density.
The authors have addressed the presence of both Fe3O4 and POM hybrids on the basis of IR, UV-Vis and magnetic measurements. How do you know that you are not obtaining heterogeneous mixtures of Fe3O4 nanoparticles and some crystals of POM hybrids? How do the authors determine the content of each component in the nanocomposite as detailed in the section 2.4.? Please, explain.
The activity of the nanocomposites for the adsorption of dyes is greater than that of the single components added to the solution separately. Do the experiment consist on the same amount of active material in both cases? How do you know?
2) The authors justify the use of transition metal ions in the active material because they can be coordinated by –C=C– and –N=N– groups from the dyes. This is not probable. Besides, when regarding to POM hybrids, they claim that “the resulting complex is difficult to be collected”. This is not true because they isolated both compounds in the solid state as single-crystals. Please, modify this piece of text.
3) Which is the role of H2O2 in the synthesis of 1 and 2? Some information is repeated in the description (e.g. Vmetanol/water = 2/3 … methanol (10 mL) and H2O (15 mL)). Please, revise the expression. The general formula given for elemental analysis should be: C H and the symbol of the remaining elements following the alphabetic order.
4) Crystallographic data in Table 1 displays more restrains than parameters for both compounds. This is not possible. What do the authors mean with “Data”? Are you referring to “reflections”? Which ones: total, unique, observed? Please indicate the Rint as well. The second values given for the R indices correspond to R2 values. Please correct this part.
5) According to the description of the crystal structures of 1 and 2, there are two and one proton in each structure, respectively. Where are the protons located? Are the phosphate groups from the POM protonated? (This is quite common). Do the authors find them in the Fourier map or do they calculate they presence on the basis of bond valence sum calculations? Thermogravimetric analysis could experimentally confirm the presence of those protons and corroborate the number of hydration water molecules per formula unit in the bulk sample.
6) The concept “intramolecular hydrogen bonds of water molecules” is completely inaccurate, because there is no such bond. The broad band in the IR spectrum is ascribed to the covalent O-H stretching vibrations.
7) To confirm the presence of the hybrid POM structures in the composite, XRD patterns in Figure 7 should be compared at least with the simulated data from single-crystal XRD analyses.
8) The molecular structure of methylene blue in Figure 8 displays one carbon atom with 5 bonds. This is unacceptable.
9) The authors observed that the activity of Fe3O4@1 increases with the pH in the 3-7 range whereas it becomes non-active above pH 9. How do you explain this fact? Is something decomposing at basic pH?
10) Please use the correct format for the references: Journal names and volume numbers need to be in italics.
11) English needs to be polished throughout the text (I am not a native speaker though). Typos, grammar mistakes and incorrect expressions are quite common all over the manuscript.
Author Response
Response to Reviewer 2 Comments
Comments and Suggestions for Authors
This manuscript by Li et al. reports on the use of polyoxometalate-hybrids@iron oxide nanocomposites for the removal of cationic dyes in aqueous solution. POM-hybrids based on Strandberg-type phosphomolybdates and organic molecules have been characterized by single-crystal X-ray diffraction, whereas the formation of composites have been analyzed by IR and UV-Vis spectroscopies, magnetic measurements and TEM images. Although results from dye absorption studies could be relevant enough to recommend the publication of this work in Nanomaterials, I cannot accept it in its current form, because there are major inconsistencies and inaccuracies all along the manuscript. Therefore, I would suggest a deep revision before reconsideration based on the following points:
Response: Thank you for your letter and comments concerning our manuscript entitled “Preparation of Fe3O4@polyoxometalates nanocomposites and their efficient adsorption of cationic dyes from aqueous solution” (ID: nanomaterials-471180). Those comments are all valuable and very helpful for revising and improving our paper, as well as the important guiding significance to our researches. We have studied comments carefully and have made correction which we hope meet with approval. Revised portion are marked in red in the paper. The main corrections in the paper and the responds to the reviewer’s comments are as follows:
1) The nature of the composite is my main concern. Are they core-shell particles? I do not think so, because TEM images show homogeneous phases of Fe3O4. There is no presence of darker points corresponding to POM units with higher electronic density.
The authors have addressed the presence of both Fe3O4 and POM hybrids on the basis of IR, UV-Vis and magnetic measurements. How do you know that you are not obtaining heterogeneous mixtures of Fe3O4 nanoparticles and some crystals of POM hybrids? How do the authors determine the content of each component in the nanocomposite as detailed in the section 2.4.? Please, explain.
The activity of the nanocomposites for the adsorption of dyes is greater than that of the single components added to the solution separately. Do the experiment consist on the same amount of active material in both cases? How do you know?
Response 1: Thank you for your question. According to a series of characterization and the relevant literatures [R1–R3], we speculated that the composites might be core-shell structure. Thermogravimetric analyses of Fe3O4, 1, 2, Fe3O4@1 and Fe3O4@2 were performed under a nitrogen flow (Fig. R1a, b). It turns out that the weight ratio of 1 and 2 in the synthesized nanocomposite is 96.68% and 96.50%, respectively. It should be noted that the loss of Fe3O4 might be attributed to the loss of its surfactant attached during the synthesis process. Meantime, the aggregation and dispersion experiments show that there is no compound left after the two nanocomposites are collected by a magnet, which indicates that the polyoxometalates hybrids and Fe3O4 are successfully combined together as the hybrids 1 and 2 are nonmagnetic. Therefore, according to the weight ratio of the nanocomposites, we tested the activity of the nanocomposites for the adsorption of dyes, as well as the single components.
[R1]Neyman, A.; Meshi, L.; Zeiri, L.; Weinstock, I.A. Direct imaging of the ligand monolayer on an anion-protected metal nanoparticle through cryogenic trapping of its solution-state structure. J. Am. Chem. Soc. 2008, 130, 16480–16481.
[R2]Fang, N.; Ji, Y.M.; Li, C.Y., Wu, Y.Y.; Ma, C.G.; Liu, H.L., Li, M.X. Synthesis and adsorption properties of [Cu(L)2(H2O)]H2[Cu(L)2(P2Mo5O23)]•4H2O/Fe3O4 nanocomposites. RSC Adv. 2017, 7, 25325–25333.
[R3]Zhang, G.J.; Keita, B.; Dolbecq, A,; Mialane, P.; Sécheresse F.; Miserque, F.; Nadjo, L. Green chemistry-type one-step synthesis of silver nanostructures based on MoV–MoVI mixed-valence polyoxometalates. Chem. Mater. 2007, 19, 5821–5823.
Fig. R1. Thermogravimetric analyses of Fe3O4, 1, 2, Fe3O4@1 and Fe3O4@2
2) The authors justify the use of transition metal ions in the active material because they can be coordinated by –C=C– and –N=N– groups from the dyes. This is not probable. Besides, when regarding to POM hybrids, they claim that “the resulting complex is difficult to be collected”. This is not true because they isolated both compounds in the solid state as single-crystals. Please, modify this piece of text.
Response 2: Thank you for your pertinent suggestion. According to your constructive comments, we have tried our best to improve the quality of our manuscript. The inappropriate sentences have been revised in the manuscript. However, it should be noted that transition metal ions are able to coordinate with most of the organic substances containing –C=C–, –N=N– and heterocyclic compounds. This is because of their lenience in change of the oxidation state and presence of unpaired electrons the metal ions react readily with molecular oxygen, thereby mediating oxygenation of other compounds easily [R4,R5].
[R4]Feng, X.H.; Tu, J.F.; Ding, S.M.; Wu, F. Photodegradation of 17β-estradiol in water by UV–vis/Fe(III)/H2O2 system. J. Hazardous Mater. 2005, 127, 129–133.
[R5]Reddy, N.B.G.; Krishna, P.M.; Kottam, N. Novelmetal-organic photocatalysts: synthesis, characterization and decomposition of organic dyes. Spectrochim. Acta A. 2015, 137, 371–377.
3) Which is the role of H2O2 in the synthesis of 1 and 2? Some information is repeated in the description (e.g. Vmethanol/Vwater = 2/3 … methanol (10 mL) and H2O (15 mL)). Please, revise the expression. The general formula given for elemental analysis should be: C H and the symbol of the remaining elements following the alphabetic order.
Response 3: Thank you for your pertinent advice. As suggested, the incorrect expressions have been revised. The order of symbol of the elementals analysis has been revised as well. Meantime, in order to explore the role of H2O2, we repeated the synthesis experiments of hybrids 1 and 2 in the absence of H2O2, respectively. It turns out that hybrids 1 and 2 still could be prepared without the addition of H2O2. Therefore, we removed the H2O2 from the article.
4) Crystallographic data in Table 1 displays more restrains than parameters for both compounds. This is not possible. What do the authors mean with “Data”? Are you referring to “reflections”? Which ones: total, unique, observed? Please indicate the Rint as well. The second values given for the R indices correspond to R2 values. Please correct this part.
Response 4: Thank you for your pertinent suggestion. We have made appropriate changes in the revised manuscript.
5) According to the description of the crystal structures of 1 and 2, there are two and one proton in each structure, respectively. Where are the protons located? Are the phosphate groups from the POM protonated? (This is quite common). Do the authors find them in the Fourier map or do they calculate they presence on the basis of bond valence sum calculations? Thermogravimetric analysis could experimentally confirm the presence of those protons and corroborate the number of hydration water molecules per formula unit in the bulk sample.
Response 5: Thank you for your question. Valence band calculation indicated that there is a one proton in {P2Mo5} of each composite. In order to balance the charge protons, the other proton in hybrid 1 was added.
6) The concept “intramolecular hydrogen bonds of water molecules” is completely inaccurate, because there is no such bond. The broad band in the IR spectrum is ascribed to the covalent O-H stretching vibrations.
Response 6: Thank you for your pertinent suggestion. We have made appropriate changes in the revised manuscript.
7) To confirm the presence of the hybrid POM structures in the composite, XRD patterns in Figure 7 should be compared at least with the simulated data from single-crystal XRD analyses.
Response 7: Thank you for your good suggestion. The simulated data from single-crystal XRD analyses of hybrids 1 and 2 have been added to Figure 7. It turns out that the presence of hybrids 1 and 2 in the nanocomposites.
8) The molecular structure of methylene blue in Figure 8 displays one carbon atom with 5 bonds. This is unacceptable.
Response 8: Thank you for your pertinent suggestion. The molecular structure of methylene blue in Figure 8 has been redrawn.
9) The authors observed that the activity of Fe3O4@1 increases with the pH in the 3-7 range whereas it becomes non-active above pH 9. How do you explain this fact? Is something decomposing at basic pH?
Response 9: Thank you for your question. As described in the article, the active site of adsorption reaction of Fe3O4@1 might concentrate on polyoxometalate hybrid 1. However, 1 in Fe3O4@1 will decompose at alkaline environment. Therefore, Fe3O4@1 becomes non-active above pH 9.
10) Please use the correct format for the references: Journal names and volume numbers need to be in italics.
Response 10: Thank you for your pertinent suggestion. The Journal names and volume numbers have been revised to italics.
11) English needs to be polished throughout the text (I am not a native speaker though). Typos, grammar mistakes and incorrect expressions are quite common all over the manuscript.
Response 11: Thank you for your good suggestion. We have tried our best and asked an English speaker to polish the language in the revised manuscript. These changes have been marked in red in revised paper. We appreciate for Editors/Reviewers’ warm work earnestly, and hope that the correction will meet with approval.

Round 2
Reviewer 2 Report
Authors have thoroughly respond to most of the reviewer’s comments. However, some points still need to be conveniently addressed:
1) If composites are formed by Fe3O4 cores and POM-hybrid shells, the presence of Mo in the particles should be confirmed EDX. I do not see a direct connection between this work and the references given in the response (R1 to R3), except for R2, which is a similar work carried out by the same authors on related POM hybrid-iron oxide composites for the removal of dyes from water. Please add this citation.
[R2] Fang, N.; Ji, Y.M.; Li, C.Y., Wu, Y.Y.; Ma, C.G.; Liu, H.L., Li, M.X. Synthesis and adsorption properties of [Cu(L)2(H2O)]H2[Cu(L)2(P2Mo5O23)]•4H2O/Fe3O4 nanocomposites. RSC Adv. 2017, 7, 25325–25333.
It would also be convenient to include the thermogravimetric analyses as supplementary information and a short discussion within the main text.
3) The general formula given for elemental analyses should be: C, H and the symbol of the remaining elements following the alphabetic order.
4) Crystallographic table: What do the authors mean with “Data”? Are you referring to “reflections”? Which ones: total, unique, observed? Please indicate all of them, and the Rint and goodness of fit (S) parameters as well. The second values given for the R indices correspond to R2 values. Please correct.
Besides, please indicate the origin of the 363 restrains used for the structure of 1.
5) Response: “Valence bond calculations indicated that there is a one proton in {P2Mo5} of each composite”. Which POM oxygen atom is protonated? it usually belongs to the phosphate group… Where is the second proton in 1? Thermogravimetric analysis on compounds 1 and 2 could experimentally confirm the presence of those protons and corroborate the number of hydration water molecules per formula unit in the bulk sample.
9) As stated by the authors “However, 1 in Fe3O4@1 will decompose at alkaline environment. Therefore, Fe3O4@1 becomes non-active above pH 9.” Please add this information in the manuscript.
10) Please revise the spelling of author’s list in reference 30.
11) Although it has improved considerably, English still needs to be polished.
Author Response
Comments and Suggestions for Authors
Auvenientlythors have thoroughly respond to most of the reviewer’s comments. However, some points still need to be con addressed:
Response: Thank you for your letter and comments concerning our manuscript entitled “Preparation of Fe3O4@polyoxometalates nanocomposites and their efficient adsorption of cationic dyes from aqueous solution” (ID: nanomaterials-471180). Those comments are all valuable and very helpful for revising and improving our paper, as well as the important guiding significance to our researches. We have studied comments carefully and have made correction which we hope meet with approval. Revised portion are marked in red in the paper. The main corrections in the paper and the responds to the reviewer’s comments are as follows:
1) If composites are formed by Fe3O4 cores and POM-hybrid shells, the presence of Mo in the particles should be confirmed EDX. I do not see a direct connection between this work and the references given in the response (R1 to R3), except for R2, which is a similar work carried out by the same authors on related POM hybrid-iron oxide composites for the removal of dyes from water. Please add this citation.
[R2] Fang, N.; Ji, Y.M.; Li, C.Y., Wu, Y.Y.; Ma, C.G.; Liu, H.L., Li, M.X. Synthesis and adsorption properties of [Cu(L)2(H2O)]H2[Cu(L)2(P2Mo5O23)]•4H2O/Fe3O4 nanocomposites. RSC Adv. 2017, 7, 25325–25333.
It would also be convenient to include the thermogravimetric analyses as supplementary information and a short discussion within the main text.
Response 1: Thank you for your question. The presence of Mo in the particles can be found in the XPS spectra of Fe3O4@1 (Figure 12f). The reference RSC Adv. 2017, 7, 25325–25333 has been cited. The thermogravimetric analyses as supplementary information and a short discussion within the main text have been added as well.
3) The general formula given for elemental analyses should be: C, H and the symbol of the remaining elements following the alphabetic order.
Response 3: Thank you for your question. As suggested, the formula for elementals analysis has been revised.
4) Crystallographic table: What do the authors mean with “Data”? Are you referring to “reflections”? Which ones: total, unique, observed? Please indicate all of them, and the Rint and goodness of fit (S) parameters as well. The second values given for the R indices correspond to R2 values. Please correct. Besides, please indicate the origin of the 363 restrains used for the structure of 1.
Response 4: Thank you for your pertinent suggestion. ‘Data’ in crystallographic table refers to observed reflections. In the final crystallographic table, the numbers of total, unique, observed reflections, the Rint and goodness of fit (S) parameters are added in Table 1. Additionally, all the related R and wR residual factors have been also supplemented in Table 1. In addition, in order to remove the ADP and NDP alerts of a lot of C and N atoms, the commands of SIMU and ISOR were used in the final refinement, therefore leading to restrains in the refinement of 1.
5) Response: “Valence bond calculations indicated that there is a one proton in {P2Mo5} of each composite”. Which POM oxygen atom is protonated? it usually belongs to the phosphate group Where is the second proton in 1? Thermogravimetric analysis on compounds 1 and 2 could experimentally confirm the presence of those protons and corroborate the number of hydration water molecules per formula unit in the bulk sample.
Response 5: Thank you for your question. In 1, O(4) of {P2Mo5} is protonated. In 2, O(22) of {P2Mo5} is protonated. The bond valence sum parameters for O(4) of 1 and O(22) of 2 is 1.376 and 1.082, respectively. The second proton in 1 was determined based on the charge balance consideration [R1].
[R1] Y. Huo, D.D. Li, R. Wan, P.T. Ma, D.D. Zhang, J.Y. Niu, J.P. Wang. Synthesis and characterization of organotriphosphonate-functionalized TM-containing polyoxotungstates. RSC Adv., 2015, 5, 106077–106082.
9) As stated by the authors “However, 1 in Fe3O4@1 will decompose at alkaline environment. Therefore, Fe3O4@1 becomes non-active above pH 9.” Please add this information in the manuscript.
Response 9: Thank you for your pertinent suggestion. The related sentences have been added to the revised manuscript.
10) Please revise the spelling of author’s list in reference 30.
Response 10: Thank you for your pertinent suggestion. The spelling of author’s list in reference 30 has been revised.
11) Although it has improved considerably, English still needs to be polished.
Response 11: Thank you for your good suggestion. We have tried our best to polish the language in the revised manuscript. These changes have been marked in red in revised paper. We appreciate for reviewers’ warm work earnestly, and hope that the correction will meet with approval.

This manuscript is a resubmission of an earlier submission. The following is a list of the peer review reports and author responses from that submission.
Round 1
Reviewer 1 Report
The authors show an interesting theory anyway a more accurate study is necessary
it is necessary to quantify the amount of polyoxometalate they put on the iron oxide nanoparticles and verifying that it is bounded to the nanoparticles.
It is necessary to describe in more accurate way the procedure to test the selective capability of absorption of the dye. They need to give the concentration of Fe3O4@1 and Fe3O4@2 put in the dye solutions pH and temperature and also test with different concentration of dye and see the efect.
The nanoparticles have a mean diameter that suggest that they are still in a single domain so they suppose to be in superparamagnetic regime at 300K, how the authors explain the magnetic results?
It is also necessary to test the aggregation state of the nanocomposites with DLS.
They have to subtract the diamagnetic contribution due to the ligand from the magnetic data.
It could be interesting see also the ZFC and FC experiments to better understand the nanoparticle magnetic behavior.
It is also necessary check a leaching of the polyoxometalate after the dye has been removed. The IR analysis is only qualitative I suggest an ICP analysis.
Please insert the number of particles counted for the size statistic, moreover make a more accurate comment to the fact that the XRD gives the same size of the TEM analysis.
It is not clear which is the advantage of inserting Ni in the complex. I suggest also a comparative work with recent work just published by the author
Reviewer 2 Report
This paper describes the preparation of Fe3O4@polyoxometalates nanocomposites and the application of these compounds in the adsorption of cationic dyes
After carefully analyzing the work in my opinion this should not be published in nanomaterials, at least in the present state.
Some of the questions that must be answered are:
In the adsorption measurements it is indicated that 1.4 10-2 mmol is used. How is this value calculated? Would it not be better to indicate the mg of adsorbent used?
Also, 15 mL are used as total volume for adsorption and it is indicated that 4 mL is taken, this amount is centrifuged for 3 min and analyzed by a spectrophotometer. Several questions: 1) How the measurement of time is made?, because from the time it is adsorbed until it is measured an important delay happen 2) The 4 mL that are used, this are incorporated into the solution again. 3) Why is it centrifuged? Does not it seem more logical to separate it with a magnet?
How have absorption spectra been performed in liquid or solid?
In magnetization curves, it is indicated that there is a large decrease in magnetization due to the non-magnetic contribution of the polyoxometalate. But if we knew at least approximately the amount of polyoxometalate that covers the nanoparticles, we could calculate the magnetization in emu / gram of magnetite and be able to compare them better.
In figure 6 c and f it would be very convenient to have both at the same magnification. In these figures the coating of polyoxometalate that the authors seem to see is not appreciated either. On the other hand, it is rather strange that 50 mg of polyoxametalate and only 7.5 mg of magnetite nanoparticles are used for the formation of the composite. This solution is mixed in 10 mL and it is 10 hours sonicating. Is the solution thermostated? In general, the prolonged use of the sonication process produces a considerable increase in the temperature of the medium.
How has the Scherrer equation been applied? That is, a single peak has been taken or an average of all the peaks of the diffractogram has been made.
The amounts used to calculate the adsorption kinetics or the adsorption are really very low and it is not indicated how much of the adsorbent is used
The authors say that it is demonstrated that the strong adsorption of these compounds is due to the electron transfer and chemical adsoption. This sentence must be carefully explained
These are some of the issues, but the paper needs to discuss the reasons why the nanocomposite adsorbs more than the polyoxometalate. If the nanoparticles are covered by the compound, the dye will be adsorbed on it, then what is the effect that they are supported on magnetite?
Reviewer 3 Report
Comments:
The manuscript entitled “Preparation of Fe3O4@Polyoxometalates nanocomposites and their efficient adsorption of cationic dyes from aqueous solution” reported the synthesis of magnetite based polyoxometalate nanocomposites for the removal of cationic dyes namely methylene blue (MB) and methyl orange (MO) from the water body through adsorption. The work has some importance from the perspective of wastewater treatment as the presence of those dyes can cause some negative consequences to human health and the environment. However, the overall quality of the work presented in this manuscript does not meet the standard of the Journal. So, my recommendation is that the submitted article needs to undergo a major revision before it can be considered for publication in the journal of Nanomaterials. My suggestions and queries are as follows-
· The abstract of the manuscript is poorly written. It only provided some qualitative discussion but no quantitative information such as adsorption efficiency, adsorption capacity, surface area of the synthesized nanocomposites etc. Must be improved.
· In the abstract, there is no data or information found about the adsorption of methyl orange (MO), a target dye that the authors studied for removal in the work (line 54).
· The authors did not clarify why they synthesized two different nanocomposites for the removal of the dyes.
· Some of the very common and important studies were missing in the submitted manuscript. E.g., adsorption isotherm, effect of pH, effect of co-existing ions etc.
· In line 14, the authors claimed the selective adsorption properties of the adsorbents but did not clearly articulate or presented evidence about how the adsorbents selectively captured only the target dye(s).
· There is no enough evidence of the stability of magnetite (Fe3O4) after its binding with polyoxometalates and after adsorption of dyes. Since magnetite is prone to auto-oxidation, it is critical to confirm its retention of original form before and after adsorption (e.g., through XPS).
· Needs to paraphrase or correct the structure of the sentences in number of places to convey the appropriate message or meaning. E. g., line 26, 32, 230, 238 etc.
· The authors did not do any comparison with other adsorbents that have already been studied for the removal of same dyes from wastewater.
· Needs to provide sufficient data to validate the reproducibility of the work (e.g., triplicate measurement).
· What will be the ultimate fate of the dyes after its adsorption on the adsorbent surface?